# Hepatoprotective Effect of Melatonin in Toxic Liver Injury in Rats

**DOI:** 10.3390/medicina55060304

**Published:** 2019-06-24

**Authors:** Oleksandra Oleshchuk, Yana Ivankiv, Halina Falfushynska, Alla Mudra, Nataliya Lisnychuk

**Affiliations:** 1Department of Pharmacology and Clinical Pharmacology, I. Horbachevsky Ternopil State Medical University, Maidan Voli sq. 1, 46001 Ternopil, Ukraine; oleshchuk@tdmu.edu.ua; 2Department of orthopedagogy and physical therapy, Ternopil V. Hnatiuk National Pedagogical University, Maxym Kryvonis str. 2, 46027 Ternopil, Ukraine; falfushynska@tnpu.edu.ua; 3Department of Medical Biochemistry, I. Horbachevsky Ternopil State Medical University, Maidan Voli sq 1., 46001 Ternopil, Ukraine; mudra@tdmu.edu.ua; 4Central Scientific Research Laboratory, I. Horbachevsky Ternopil State Medical University, Maidan Voli sq. 1, 46001 Ternopil, Ukraine; lisnychuk@tdmu.edu.ua

**Keywords:** rat, melatonin, liver, lipoperoxidation, antioxidant, nitric oxide

## Abstract

*Background and objectives:* toxic liver injury results in nitrooxidative stress. Melatonin is a potent free radical scavenger, an inducible nitric oxide synthase (iNOS) inhibitor and an activator of antioxidant enzymes. The aim of this study was to investigate the hepatoprotective effect of exogenous melatonin on animals with acute toxic hepatitis. *Material and methods:* 36 healthy Sprague-Dawley male rats were split into three equal groups and given carbon tetrachloride (CCl_4_), 2 g/kg (CCl_4_ group) or the same dose of CCl_4_ and melatonin, 10 mg/kg (CCl_4_/melatonin group) or saline (control group). The effect of melatonin on prooxidant and antioxidant system indexes, NO and NOS levels in serum and liver, data of mitochondrial chain functions and cytolysis in liver were evaluated in all three groups. *Results:* melatonin significantly decreased activities of AST, ALT, ceruloplasmine and thiobarbituric acid reactive substance (TBARS) in serum. Catalase activity was lowered in serum but not in the liver. Hepatic TBARS, lipid hydroperoxides and glutathione concentrations were decreased, while superoxide dismutase, mitochondrial cytochrome oxidase and succinate dehydrogenase activities increased. Melatonin inhibited synthesis of stable NO metabolites in serum: NO_2_-by 37.9%; NO_3_-by 29.2%. There was no significant difference in content NO_2_-in the liver, but concentration of NO_3_-increased by 32.6%. Melatonin significantly reduced iNOS concentrations both in serum (59.7%) and liver (57.8%) but did not affect endothelial isoform enzyme activities neither in serum, nor in liver. The histopathological liver lesions observed in the CCl_4_/melatonin group were less severe than those seen in the CCl_4_ group. *Conclusions:* we demonstrated an ameliorating effect of melatonin on prooxidants and antioxidants, NO-NOS systems balance, mitochondrial function and histopathological lesions in the liver in rats with CCl_4_-induced hepatitis.

## 1. Introduction

Melatonin is a secretory product of the vertebrate pineal. Once synthesized, melatonin is not stored in pineal cells, but is quickly released into the bloodstream [1]. Pineal melatonin plays an important role in sleep cycles regulation (i.e., circadian rhythm). Its production is influenced by detection of light and darkness. Melatonin peaks during the nighttime hours and induces physiological changes that promote sleep, such as decreased body temperature and respiratory rate. Melatonin is also produced in multiple cells and organs [2]. In contrast to the pineal melatonin, the one generated in other organs seems to have paracoid, autocoid, immunomodulative and antioxidant local activities [1,3]. Melatonin acts directly on target tissues through specific binding sites which are situated in the plasma membrane and nucleus of cells. The specific binding sites of melatonin are MT_1_ and MT_2_ membrane receptors belonging to the G-protein coupled receptor family [4,5].

It is known that melatonin is involved in antioxidant defense system protecting molecules from oxidative damage. Melatonin is a potent free radical scavenger, and additionally stimulates a number of antioxidant enzymes. Melatonin can enter each cell and may carry out its antioxidant activity with equal efficiency in a variety of cellular compartments, i.e., in the nucleus, cytosol and membranes. Both in vitro and in vivo studies suggest that melatonin’s scavenging activity on highly toxic hydroxyl radical and other oxygen centered radicals is not mediated by receptor mechanisms [1,3]. Melatonin is found to be more effective in protecting against oxidative damage than any other known antioxidants (e.g., mannitol, glutathione, and vitamin E) [6]. Therefore, it may provide protection against degenerative and proliferative injures. The protection seemed to be due to shielding macromolecules, particularly DNA, from such impairs [7]. However, these antioxidant effects require concentrations of melatonin that are much higher than its peak nighttime serum level. Thus, an antioxidant effect of melatonin in humans probably occur only at pharmacologic concentrations [7].

Nitric oxide (NO) is an important messenger regulating nervous, immune, and cardiovascular homeostasis [8]. High level of NO contributes in toxic liver injury [9,10]. Inducible isoform of NO synthase (iNOS) is responsible for overproduction of NO [11]. NO can also react with superoxide anion (O_2_·^−^), leading to formation the peroxynitrite anion (ONOO^−^), which oxidizes sulfhydryl groups and generates hydroxyl radical (·OH) [8].

Besides the direct scavenging of free radicals, melatonin influences the oxidative stresses status in an indirect way. It is able to stabilize the inner mitochondrial membrane, which improves the electron transport chain located there [7]. Melatonin inhibits iNOS that are no any longer beneficial to the system, but increase oxidative stress due to its conversion together with reactive oxygen species (ROS) into the destruction of reactive nitrogen species [7,9]. Melatonin, both in pharmacological and in physiological concentrations, increases activity of endogenous antioxidant enzymes such as glutathione peroxidase, superoxide dismutase and catalase, which are important factors in maintaining liver structures and functions [6,8]. Moreover, melatonin prevents hepatocellular necrosis due to containment of the pronecrotic oxygen radical load, observed as inhibition of the lipid peroxidation and hydrogen peroxide increase in the liver [12]. Melatonin can decrease the levels of ROS, which is associated with the development and progression of cancer and several other disease states, and also prevents methotrexate-induced hepatorenal oxidative injury by reducing the levels of malondialdehyde and the activity of myeloperoxidase and increasing glutathione levels [13,14].

The aim of this study is to investigate hepatoprotective effects of melatonin in an experimental toxic in rat’s liver injury.

## 2. Materials and Methods

### 2.1. Animal Model

Thirty-six male Sprague-Dawley rats (Ternopil State Medical University vivarium, Ukraine) 8–10-week-old, 180–220 g weight were used in these experiments. The animals were starved for 12 h before experiments and allowed water ad libitum. All the animals received care in compliance with the Guide for the Care and Use of Laboratory Animals (National Institutes of Health Publication No. 85–23, revised 1985). Experiments were performed in accordance with the National Institutes of Health Guide for the Care and Use of Laboratory Animals and were approved by the local animal committee (Minutes No 40 dated 15 March 2017), there were no violations of ethical guidelines during carrying out this research.

### 2.2. Induction of Toxic Liver Injury

The experimental animals were split into three groups. The first group received carbon tetrachloride (CCl_4_), 2 g/kg in 50% oil solution of olive oil. The second one received the same dose of CCl_4_ and melatonin (Sigma-Aldrich Chemie GmbH, Schnelldorf, Germany), 10 mg/kg. The third group received saline as control. All the injections were administered intraperitoneally and were repeated for three consecutive days at 11 a.m. At the fourth day animals were anesthetized with ketamine (75 mg/kg). All the animals were sacrificed by exsanguination.

### 2.3. Blood and Tissue Acquisition

Blood samples were obtained from the right ventricle via left anterior thoracotomy at the time of the sacrifice of the animal. Blood was collected with a sterile syringe without anticoagulant and centrifuged at 2000× *g* to separate the serum. The serum samples were stored at −20 °C until use for AST and ALT assays. For cytokine and NOS determination serum was gained from blood samples by clotting for 2 h on ice; then the serum was centrifuged at 2500× *g* (4 °C), filtered, aliguoted, and frozen at −20 °C until assayed for TNF-α, IL-1β, IL-6, eNOS, and iNOS. A liver sample of 1 g from a left lateral lobe from each animal was frozen immediately after acquisition and stored in liquid nitrogen until used for eNOS, and iNOS assays. 

### 2.4. Determination of Liver Enzymes and Urea in Serum

Determination of AST, ALT, and urea in serum is performed with the Raytman-Frenkel method, using standard kits (Filisit-diagnostic, Ukraine) according to the manufacturer’s instruction. The activities of AST and ALT in serum were expressed in mmoL/(L × h), and urea concentration in mmoL/L.

### 2.5. NOS Assays

Expression of eNOS and iNOS are investigated in serum and liver with ELISA method, using «Enzyme-linked Immunosorbent Assay Kit for Rat Nitric Oxide Synthase 3, Endothelial (NOS3)», USCN, Life Science Inc., E90868Ra and «Enzyme-linked Immunosorbent Assay Kit for Rat Nitric Oxide Synthase 2, Inducible (NOS2)», USCN, Life Science Inc, E90837Ra respectively, according to the manufacturer’s protocol. eNOS and iNOS activities in serum were expressed in U/mL and in hepatocytes expressed as U/g. The test was carried out immediately after acquisition of the samples, or the samples were frozen at −20 °C if the test execution was postponed.

The procedure of liver cells lysis was performed as follows:Liver homogenates were prepared using saline at a ratio of 1:10.Liver cells suspension was centrifuged for 5 min at 300 g, then the supernatant was removed.Cells were washed twice in saline, after each wash they were centrifuged for 5 min at 300 gLysis Phosphate-buffered saline was added in correlation 1 mL of buffer at 1 × 106 liver cells. Suspension was centrifuged for 5 min at 300 g.The supernatant was collected. The enzymes activities were estimated immediately or frozen at −20 °C for postponed testing.

### 2.6. Oxygen Reactive Substances and Antioxidant Enzymes Activities

The concentration of thiobarbituric acid reactive substance (TBARS) [15], nitrite (NO_2_^-^) and nitrate (NO_3_^−^) anions [16], catalase (CAT) activities were determined in plasma and liver [17], antioxidant protein ceruloplasmine (CP) concentration only in blood [18], superoxide dismutase (SOD) activity [19], lipid hydroperoxides (LHP) [20], mitochondrial enzymes activity cytochrome oxidase (CHO) [21] and succinate dehydrogenase (SDH) [22], concentration of sulfhydril group (GSH) [23] only in liver.

### 2.7. Histopathological Study

Liver scraps were fixed in 10% neutral-buffered formalin solution for five days, embedded in paraffin and sectioned. The scraps were stained with hematoxylin and eosin.

### 2.8. Statistical Analysis

Data was tested for normality and homogeneity of variance using Kolmogorov-Smirnoff and Levene tests, respectively. For the data that was not normally distributed, non-parametric tests (Kruskall-Wallis ANOVA or Mann-Whitney U-test) were performed. Normalized, Box-Cox transformed data were subjected to the principal component analysis (PCA) with NIPALS algorithm to differentiate individual specimens by the set of their indices. The classification tree based on all studied traits was built with Classification and Regression Tree (CART) software using raw (non-transformed) data. All statistical calculations were performed with Statistica v. 10.0 and Excel for Windows-2010. Differences were considered significant if the probability of Type I error was less than 0.05. The data are presented as means (M) ± standard deviation (SD) unless indicated otherwise.

## 3. Results

The development of cytolysis in the liver in the CCl_4_-group is proved with increased activities of ALT and AST in the serum. An activation of free radical processes in acute toxic hepatitis is confirmed by the increase of LHP and TBARS liver concentrations (Table 1).

We have demonstrated that acute toxic hepatitis caused a significant increase in serum stable nitric oxide metabolite content NO_2_^−^.

Simultaneously, a significant decrease of antioxidant enzymes activities in the liver is found in the CCl_4_ group. Activities of SOD and CAT are reduced. Blood catalase activity and ceruloplasmine concentration are increased comparing to control. Depletion of the glutathione (GSH) pool in CCl_4_ liver is proved by statistically significant decreases of its amount. The content of urea in serum was significantly reduced compared to the control (Table 1). 

Activities of mitochondrial enzymes SDH and CHO in hepatitis are significantly decreased compared to the control (Table 1). In toxic hepatitis, melatonin causes a significant inhibition of nitric oxide synthesis in the blood, as shown in Table 2.

Melatonin dosing dose not result in statistically significant change in liver NO_2_^-^ concentrations in the CCl_4_-group as compared to the CCl_4_/melatonin-group. However, the concentration of nitrate-anion is statistically significantly increased in the CCl_4_-group.

There are no significant differences in concentrations of endothelial NO-synthase isoforms in neither the blood, nor in the liver, with toxic lesions of CCl_4_ compared to the CCl_4_/melatonin-group of animals. Anyway, melatonin reduced the concentration of inducible NOS in the serum and in the liver significantly (Table 3).

PCA showed that 78.7% of variation in the studied traits was associated with two principal components (Factors 1 and 2) (Figure 1). The first principal component explained how 69.9% of the total variation had high loadings (~0.6 or higher) of stress parameters in the liver (eNOS, iNOS, NO_3_, TBARS, SOD, catalase, ALT, AST, SDH, CHO, GSH), as well as eNOS, iNOS, NO_2_, TBARS, catalase and urea in the blood. The second principal component had high loadings of NO_3_ in the blood.

Control and CCl_4_-groups were mostly associated with Factor 1 with opposite loading one by one. The CCl_4_/melatonin group had loaded on Factor 2. The CCl_4_/melatonin group had interim location with control and the hepatitis-exposed one.

The most prominent parameters characterized of control group were eNOS, catalase activity and nitrite ion level in the liver and urea in blood samples. A set of animals from the CCl_4_ group were separated by the indexes of nitrite ion in blood, ALT in liver and TBARS and iNOS in both liver and blood. No indices were attached significantly to the CCl_4_/melatonin group. 

The liver degenerative and lobular necrosis center of the hepatocyte with lympho-histiocytic infiltrates were found in CCl_4_-group (Figure 2).

It has been demonstrated that structural organization of liver lobules is broken. Central veins are expanded but do not contain erythrocytes. Sinusoids are hardly identified and contained an increased number of macrophages. Portal vessels are moderately expanded, and perivascular spaces are thickly infiltrated with lymphocytes and histiocytes.

Hepatocytes are only slightly damaged in the periphery of liver lobules. The degenerative lesions are mainly expressed in the center of the lobular cells. They are characterized by destruction of cytoplasm or granular savings; some of the cells do not contain nuclei. The development of protein and fine drops of fatty dystrophies that transit to necrosis was determined. Portal tract was infiltrated by histiocytes and leukocytes. Portal tracts vessels were increased in size 

ALT activity in CCl_4_/melatonin group was lower by 33.3% in comparison with the CCl_4_ group and remained significantly higher than the control. The AST activity was decreased by 29.5% compared CCl_4_ group. The content of urea in serum CCl_4_/melatonin group CCl_4_ did not significantly change compared to the CCl_4_ group of animals (Table 1). 

In CCl_4_-induced hepatitis, melatonin dosing results in decrease of TBARS concentrations both in the liver and in serum for CCl_4_ (Table 1), respectively, as well as decrease of LHP concentration in the liver.

In addition, melatonin prevents an inhibition of mitochondrial respiration, which occurred in the CCl_4_ injury. Therefore, the activity of mitochondrial enzymes such as SDH and CHO increased in the liver.

We have found that the trabecular structure of hepatic lobules was partially preserved in CCl_4_/melatonin animals (Figure 3). The central veins are slightly dilated and filled with blood; a perivascular space has scant lympho-histiocytic infiltrates. Sinusoids contained a small number of macrophages. The portal tracts have moderate perivascular cell infiltration. Hepatocytes in general do not present severe structural changes; however, protein dystrophy is demonstrated in some cells. Necrosis are identified rarely.

## 4. Discussion

Carbon tetrachloride (CCl_4_)-induced liver injury is a classic model of chemical liver injury. CCl_4_ is a potent hepatotoxin associated with histopathological effects of inflammatory or oxidative stress and cell death [24]. The program of the Boston Medical Center has the purpose of providing a designation of containment levels that outline the requisite administrative controls, engineering controls and personal protective equipment necessary to protect researchers, and the environment from potential exposures involving animals carbon tetrachloride that have chemical containment level 2 [25]. It is suitable for work involving the use of chemicals, which can pose a moderate risk to personnel and the environment from an animal exposed to a hazardous substance [25]. 

Using a classical model of acute toxic liver damage by tetrachloromethane, we have found a significant increase in AST and ALT activity, which indicates the development of breach membrane integrity of hepatocytes and their organelles and exit enzymes in the intercellular environment. The molecular mechanisms of membrane damaging action by CCl_4_ in vivo should distinguish between the following factors: (1) the binding of molecules xenobiotics with hydrophobic areas of biomembranes; (2) the ability to metabolize xenobiotic with the participation of specific enzyme system monooxygenases of endoplasmic reticulum to form highly reactive metabolites that interact with the bio structures of cells [26,27]. One of these metabolites can be nitric oxide and free radical products of its metabolism. The level of stable metabolites of NO nitrite and nitrate anions in the affected organ with CCl_4_ hepatitis were decreased. Similar results were obtained in studies N. Tanaka et al. (1999) [26]. These results are correlated with the content of NOS isoforms. It was established that the content of inducible form of the enzyme on the third day of the experiment increases in the liver and in the blood. However, endothelial form was decreased. The increase in production of iNOS is stimulated by formation of NO in the cells of the liver and toxic damage is necessary to ensure the phenomenon of “working hyperemia.” The expansion of the vessel is targeted at the redistribution of oxygen and plastic resources to restore the disturbed functions [10]. An activation of free radical processes with the increase of lipid hydroperoxides and TBA-active products are key components of the pathogenesis in toxic liver disease [26,27].

This study shows that the enzymatic activity of the antioxidant system of the liver is decreased significantly under the influence of carbon tetrachloride, which is consistent with the results of other investigators [28,29]. As the catalase is the intra-cellular enzyme, the increase of its activity in plasma and decrease in the liver can be interpreted as a result of the integrity or permeability of plasma membrane and the exit of its enzyme in the extracellular space. The decreasing of mitochondrial enzyme activity and SDG, CHO, that indicates a violation of mitochondrial respiration in rats with CCl_4_ hepatitis, were observed. According to several researchers and based on the results of our experiments, hepatosis, caused by a single injection of CCl_4_, is characterized by the type of anaerobic metabolism (the ability to maintain vital functions was activated by glycolysis), weak breathing in vivo, stimulated by hypoxic states of affected hepatocytes. Intensively absorbing the glucose, they cannot practically utilize substrates of the Krebs cycle, which are the main substrates of the tissue respiration. Mitochondria in these cells is altered and demonstrated a weaker phosphate activity [30]. 

The morphological pattern is typical for acute toxic hepatitis, namely the development of dystrophic and necrotic lesions in the centre of the lobular hepatocytes. Consequently, this results in disruption of structural organization and severe acute inflammatory reaction in the periportal areas. Therefore, the application of melatonin in rats with toxic hepatitis marked the decrease of liver injuring manifestations that was proved histologically. It can be explained by both its direct and indirect antioxidant action, modulation of NO-system and influencing on mitochondrial function and process of cytolisis, which we achieved in our experiments [5,31]. Antioxidant properties of melatonin are clearly recognized, and they have been studied extensively during last decades [3,32]. The properties of melatonin as a free radicals scavenger, according to Cabeza et al. (2001) [27], is caused by the inactivation of superoxide anion, possibly through xanthine oxidase way. Melatonin also inhibits the formation of peroxynitrite, a toxic oxidants, which formed with the participation of nitric oxide and superoxide anion in terms of pathology [33]. The effect of melatonin on lipid peroxidation, antioxidant enzymes and glutathione content at healthy rats was depicted in our previous work [34]. It was shown that the introduction of the agent promotes the inhibition of lipid peroxidation processes; enhance antioxidant protecting the body and link mitochondrial enzymes chain. Analogical experimental data was shown in case of melatonin introduction against ischemia-reperfusion injury in rats [35]. Melatonin plays a role in the prevention of oxidative damage. The results of our study confirmed the antioxidant effect of melatonin by reducing oxidative stress in toxic hepatitis. We found that melatonin treatment slightly increased activities and/or level of antioxidant parameters, among them SOD, catalase and glutathione significantly reduced oxidative lesions named TBARS and lipids hydroperoxides when compared with the CCl_4_-treated animals with hepatitis injury. The balanced organisms’ oxidative stress system response against studied treatments described by multiply regression equation: TBARS (liver) = 5.89 + 0.07 × SOD − 0.23 × CAT + 0.28 × CHO − 1.11 × GSH, R2 = 0.85, F (4,13) = 19.125, *p* < 0.001. Nevertheless, melatonin affect only partially ameliorated CCl_4_/melatonin group of animals and was located in the mid of control and hepatitis one (Figure 1). This fact needs detail evaluation and mechanism findings. However, due to our knowledge, some studies reported no-effect or abolished melatonin effect in different species and vary pathology [36]. The studies have shown that melatonin inhibits the activity of inducible NO synthase, beside its NO and peroxynitrite scavenging activity [37]. This also is confirmed here. The concentration of iNOS both in the blood and the liver are decreased when exogenic melatonin is used for CCl_4_ injured rats. It has been recently proposed that in addition to the previously reported free radical scavenging cascade, melatonin is also involved in a concurrent “chelating cascade” thereby contributing to a reduction in oxidative stress, due to the copper sequestering ability of melatonin [38]. The results of the study of melatonin and its precursors, tryptophan and serotonin, for their metal binding affinities for both essential and toxic metals, showed the ability of melatonin to form complexes with copper, suggesting a further role for melatonin in the reduction of free radical generation and metal detoxification [39]. However, the potential roles for insufficient copper in the pathogenesis of liver disease, namely the Wilson’s disease, requires further study of the interaction of melatonin and copper in the reduction of free radicals [40]. Moreover, the research results also show that the change in the bioavailability of copper predicts early atherosclerosis as a major risk of cardiovascular disease, in obese patients with hepatic steatosis [41].

Our data also suggests that melatonin regulates mitochondrial homeostasis in toxic hepatitis. It is known that melatonin has a number of effects at the mitochondrial level that improve the wellbeing of cells. It influences both the electron transport chain and oxidative phosphorylation by increasing electron transport and ATP production. It also counteracts the damage resulting from the exposure of mitochondria to tertbutyl hydroperoxide, restores reduced glutathione levels and enhances the production of ATP [42]. The decrease of liver cytolysis and less severe liver lesion in melatonin treatment group aver its hepatoprotective effect. Therefore, partial restoration of the structural organization in the liver, reduction of dystropho-necrotic manifestations, was observed in case of the introduction of melatonin. Our result describes not only its receptor and antioxidant effects but also metabolic properties of melatonin protective action.

## 5. Conclusions

The study has shown that administration of melatonin for rats with toxic hepatitis leads to restoration of morpho-functional state of the liver that is indicated by:a decrease of cytolysis enzymes activity, and activation of mitochondrial respiration in hepatocytes;partial restoration of the structural organization of the liver, reduction of its dystrophic necrotic manifestations;reduction of the oxidative stress manifestations and restoration of antioxidant system balance;block of iNOS both in blood and in the liver leads to reduction of nitro oxidative stress.

## Figures and Tables

**Figure 1 medicina-55-00304-f001:**
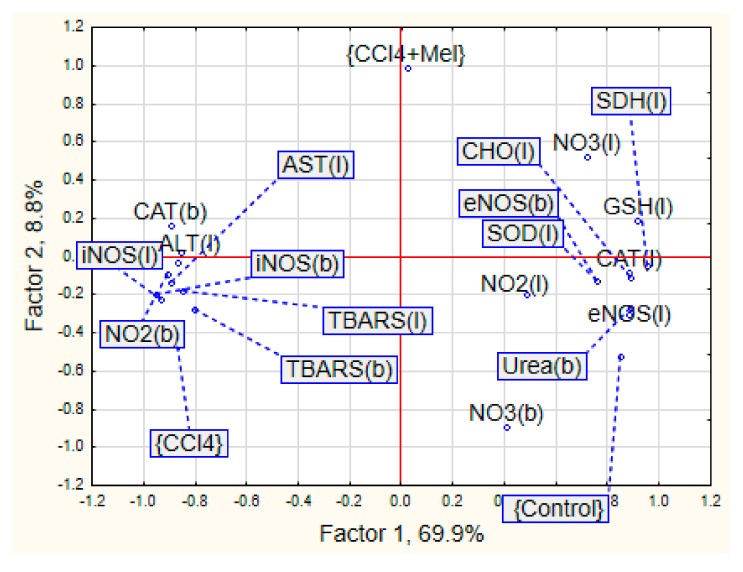
The principal component analysis of the biochemical parameters under liver injury and its repair with melatonin. b – blood; l – liver; Hep – hepatitis; Mel - melatonin; ALT - alanine aminotranferase; AST – aspartate aminotranferase; CAT – catalase; SOD – superoxide dismutase; TBARS - thiobarbaturic acid reactive substances; CHO - cytochrome oxidase; SDH – succinate dehydrogenase; NO_2_^−^ – nitrites; NO_3_^−^ – nitrates; eNOS – endothelian NO synthase; iNOS – inducible NO synthase.

**Figure 2 medicina-55-00304-f002:**
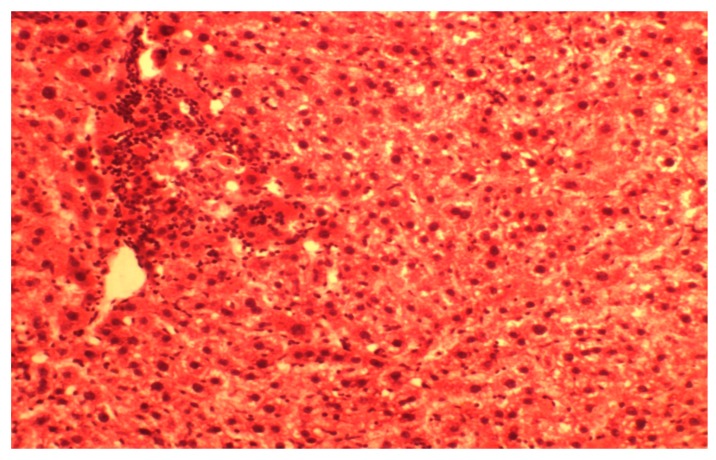
Histological structure of hepatic lobules in acute toxic liver damage in the CCl_4_ group. Dystrophy-necrotic changes of central lobular zone. Stained by hematoxylin and eosin. ×160.

**Figure 3 medicina-55-00304-f003:**
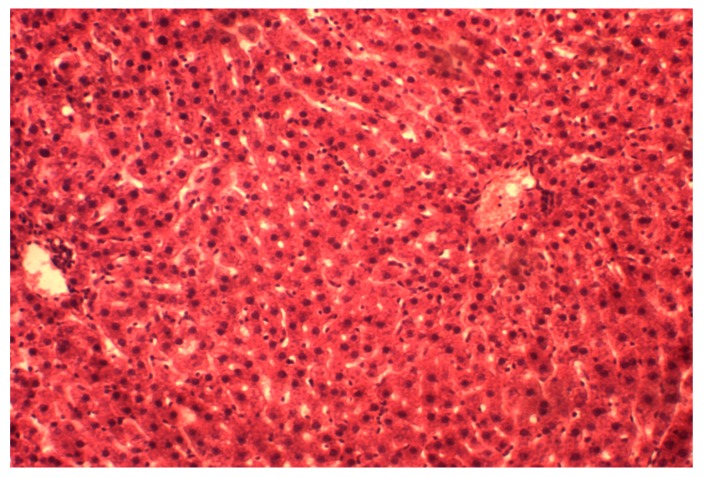
Histological structure of hepatic lobules in the CCl_4_/melatonin group. The partial restoration of trabecular structures, a phenomena of protein degeneration in hepatocytes. Stained by hematoxylin and eosin. ×160.

**Table 1 medicina-55-00304-t001:** Laboratory indicators in blood and liver.

	Control	CCl_4_	CCl_4_/Melatonin
ALT, mmoL/(g × L)	0.45 ± 0.10	1.13 ± 0.07	0.75 ± 0.04
		*p* < 0.01	*p*_1_ < 0.01
AST, mmoL/(g × L)	1.70 ± 0.21	3.17 ± 0.16	2.23 ± 0.60
		*p* < 0.01	*p*_1_ < 0.01
CAT (serum), mcat/L	14.58 ± 0.40	18.93 ± 0.30	15.84 ± 0.17
		*p* < 0.005	*p*_1_ < 0.001
CAT (liver), mcat/kg	4.31 ± 0.11	2.71 ± 0.32	3.31 ± 0.13
		*p* < 0.01	*p*_1_ > 0.05
SOD, U/g	4.59 ± 0.10	2.95 ± 0.19	3.60 ± 0.19
		*p* < 0.005	*p*_1_ ≤ 0.05
LHP, U/g	1.77 ± 0.14	3.05 ± 0.13	2.43 ± 0.21
		*p* < 0.005	*p*_1_ ≤ 0.05
TBARS (serum), mmoL/L	2.18 ± 0.11	3.18 ± 0.20	2.36 ± 0.05
		*p* < 0.01	*p*_1_ < 0.01
TBARS (liver), mmoL/kg	3.06 ± 0.10	4.40 ± 0.22	3.46 ± 0.19
		*p* < 0.01	*p*_1_ < 0.05
CP, mg/L	230.42 ± 5.35	278.54 ± 7.99	250.83 ± 12.84
		*p* < 0.01	*p*_1_ > 0.05
GSH (liver), mmoL/kg	4.20 ± 0.08	2.86 ± 0.10	3.67 ± 0.14
		*p* < 0.005	*p*_1_ < 0.01
CHO, mmoL/(kg × min)	8.73 ± 0.28	6.65 ± 0.16	7.51 ± 0.48
		*p* < 0.01	*p*_1_ < 0.01
SDH, mmoL/(kg × min)	8.85 ± 0.11	6.78 ± 0.11	7.51 ± 0.12
		*p* < 0.005	*p*_1_ < 0.001
Urea, mmoL/L	5.88 ± 0.11	3.83 ± 0.23	4.39 ± 0.14
		*p* < 0.001	*p*_1_ > 0.05

Notes: the results are presented as mean ± SD (*n* = 12); significantly differences: *p*-versus control, *p*_1_-versus CCl_4_ group. ALT – alanineaminotranferase; AST – aspartateaminotranferase; CAT – catalase; SOD – superoxidedismutase; CP – ceruloplasmine; TBARS – thiobarbaturic acid reactive substances; LHP – lipid hydroperoxides; CHO – cytochromeoxidase; SDH – succinate dehydrogenase; GSH –glutathione.

**Table 2 medicina-55-00304-t002:** Content of nitrite and nitrate anions in blood and liver.

	Blood (µmol/L)		Liver (µmol/kg)	
	NO_2_^−^	NO_3_^−^	NO_2_^−^	NO_3_^−^
Control	1.17 ± 0.06	10.21 ± 0.07	2.19 ± 0.15	8.77 ± 0.26
CCl_4_	3.18 ± 0.26	8.36 ± 0.18	1.80 ± 0.18	6.79 ± 0.24
	*p* < 0.005	*p* < 0.005	*p* > 0.05	*p* < 0.01
CCl_4_/melatonin	1.98 ± 0.17	5.92 ± 0.05	1.88 ± 0.07	9.01 ± 0.45
	*p* < 0.01	*p* < 0.005	*p* > 0.05	*p* > 0.05
	*p*_1_ < 0.01	*p*_1_ < 0.005	*p*_1_ > 0.05	*p*_1_ < 0.01

Notes: the results are presented as mean ± SD (*n* = 12); significantly differences: *p* – versus control; *p*_1_ – versus CCl_4_ group. NO_2_
^–^ nitrites; NO_3_ – nitrates.

**Table 3 medicina-55-00304-t003:** Content eNOS and iNOS in liver and blood.

	Serum		Liver (1 mL^−1^ × 10^6^ Cells)	
	eNOS U/mL	iNOS ng/mL	eNOS U/mL	iNOS ng/mL
Control	2.33 ± 0.26	15.38 ± 0.82	3.79 ± 0.17	2.30 ± 0.34
CCl_4_	1.27 ± 0.07	96.51 ± 5.21	1.26 ± 0.20	11.28 ± 0.79
	*p* < 0.05	*p* < 0.001	*p* < 0.01	*p* < 0.001
CCl_4_ + melatonin	1.72 ± 0.21	38.88 ± 3.06	2.05 ± 0.13	4.76 ± 0.34
	*p* > 0.1	*p* < 0.001	*p* < 0.01	*p* < 0.01
	*p*_1_ > 0.05	*p*_1_< 0.001	*p*_1_ > 0.05	*p*_1_ < 0.001

Notes: The results are presented as mean ± SD (*n* = 12); significantly differences: *p* – versus control; *p*_1_ – versus CCl_4_ group. eNOS – endothelian synthase NO; iNOS – inducible synthase NO.

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
