# Peer review of "Hepatoprotective Effect of Melatonin in Toxic Liver Injury in Rats"

_medicina, 2019, doi:10.3390/medicina55060304_

Round 1
Reviewer 1 Report
no comments
Author Response
Manuscript has been revised by professional interpreter, so we will improve English language and style
Reviewer 2 Report
HEPATOPROTECTIVE EFFECT OF MELATONIN has been extensively reported by earlier studies. Current study does not add anything new to existing information and hence, can not be accepted for publication.
Author Response
The innovation in our study is research of possible mechanism of hepatoprotective effect of melatonin as selective inhibitor of inducible NO synthase.
The Manuscript has been revised by professional interpreter, so we will improve English language and style
Reviewer 3 Report
The paper presented by the authors is a nice work on the hepatoprotective effect of melatonin in an induced toxic liver injury in rats, this effect of melatonin has been already studied in other works in even more complex models, and those works are not cited, for example:
J Physiol Biochem. 2014 Jun;70(2):441-50. doi: 10.1007/s13105-014-0322-7
J.Pineal Res.2003;34:282–287
Those are just a few example, I think that the introduction should be less general and more focused on the hepatoprotective effects already reported. Furthermore some novel citations on modern works on antioxidant activity of melatonin and analogues should be reported to give at least a modern touch to the paper for example:
Chem. Res. Toxicol. 2019, 32, 100−112
With a more sound and appropriate introduction I think that the paper could work. But some more specific reference are needed.
The work in general is well performe even if the novelty is quite low because the toxic damage induced is already well studied and somehow old, but anyway it can be interesting applied on Melatonin (that has already a demonstrated hepatoprotective effect), the results are clear and in a general way the paper works.
Furthermore the authors did not complete the sections on Author Contributions, Acknowledgments and Conflicts of Interest
For this reason I suggest to Accept after Major Revisions.
Author Response
The Manuscript has been revised by professional interpreter, so we will improve English language and style
The latest papers about melatonine protective effect, including that you recomend, are included in introduction and discussion in revised version
The innovation in our study is research of possible mechanism of hepatoprotective effect of melatonin as selective inhibitor of inducible NO synthase.
the sections on Author Contributions, Acknowledgments and Conflicts of Interest are complete in new version of manuscrip
Reviewer 4 Report
Manuscript ID: medicina-499451
Title: Hepatoprotective effect of melatonin in toxic liver injury in rats
This is an interesting manuscript investigating the effect of melatonin, a potent free radical scavenger, on toxic liver injury induced by nitrooxidative stress. The authors have followed up activities of releasing enzymes from liver with melatonin treatment in rats with toxic liver injury with CCl4 treatment. Then they have analyzed nitric oxide superfamily and NOS family as well.
Although, this manuscript is based on a lots of scientific evidences, a number of issues dampen the reviewer's enthusiasm. There are serious critique regarding treated chemical, especially CCl4 is not allowed to use for animals because it is very harmful. Also major revisions which should be addressed before reconsidering this manuscript for publication.
Please thoroughly edit the manuscript for typographical and formatting errors. For example, there are several instances where some references have entered journal names in abbreviations, but others do not.
Author Response
Thanhs for critical review.
The Manuscript has been revised by professional interpreter, so we will improve English language and style
Carbon tetrachloride induced hepatitis model is classic model https://onlinelibrary.wiley.com/doi/full/10.1046/j.1365-2613.2000.00144.x
https://www.mdpi.com/2072-6643/9/10/1072/htm
that is recomemded by Guideline for preclinical study in Ukraine and are used by other international researchers for preclinincal testing of drugs.
https://www.hindawi.com/journals/sci/2016/5720413/abs/
the manuscript is revised for typographical and formatting errors
Round 2
Reviewer 2 Report
Accept in present form
Author Response
Please see the word file below.

Reviewer 3 Report
The paper has been corrected and all my suggestions have been fulfilled, I suggest to accept the manuscript
Author Response
Please see the word file below.
